# A Complex Intrachromosomal Rearrangement Disrupting *IRF6* in a Family with Popliteal Pterygium and Van der Woude Syndromes

**DOI:** 10.3390/genes14040849

**Published:** 2023-03-31

**Authors:** Alya A. Al-Kurbi, Elbay Aliyev, Sana AlSa’afin, Waleed Aamer, Sasirekha Palaniswamy, Aljazi Al-Maraghi, Houda Kilani, Ammira Al-Shabeeb Akil, Mitchell A. Stotland, Khalid A. Fakhro

**Affiliations:** 1College of Health and Life Sciences, Hamad Bin Khalifa University, Doha 34110, Qatar; 2Department of Human Genetics, Sidra Medicine, Doha 26999, Qatar; 3Division of Plastic and Craniofacial Surgery, Sidra Medicine, Doha 26999, Qatar; 4Department of Surgery, Weill Cornell Medical College, Doha 24144, Qatar; 5Department of Genetic Medicine, Weill Cornell Medical College, Doha 24144, Qatar

**Keywords:** popliteal pterygium syndrome, 1q32, *IRF6* gene, Cleft palate, cleft lip, syndactyly, intrachromosomal rearrangements, whole-genome sequencing

## Abstract

Clefts of the lip and/or palate (CL/P) are considered the most common form of congenital anomalies occurring either in isolation or in association with other clinical features. Van der woude syndrome (VWS) is associated with about 2% of all CL/P cases and is further characterized by having lower lip pits. Popliteal pterygium syndrome (PPS) is a more severe form of VWS, normally characterized by orofacial clefts, lower lip pits, skin webbing, skeletal anomalies and syndactyly of toes and fingers. Both syndromes are inherited in an autosomal dominant manner, usually caused by heterozygous mutations in the Interferon Regulatory Factor 6 (*IRF6*) gene. Here we report the case of a two-generation family where the index presented with popliteal pterygium syndrome while both the father and sister had clinical features of van der woude syndrome, but without any point mutations detected by re-sequencing of known gene panels or microarray testing. Using whole genome sequencing (WGS) followed by local de novo assembly, we discover and validate a copy-neutral, 429 kb complex intra-chromosomal rearrangement in the long arm of chromosome 1, disrupting the *IRF6* gene. This variant is copy-neutral, novel against publicly available databases, and segregates in the family in an autosomal dominant pattern. This finding suggests that missing heritability in rare diseases may be due to complex genomic rearrangements that can be resolved by WGS and de novo assembly, helping deliver answers to patients where no genetic etiology was identified by other means.

## 1. Introduction

Orofacial clefts (OFC), specifically, clefts of the lip and/or the palate (CL/P), are among the most common form of congenital craniofacial anomalies affecting about 1 in 500 to 1 in 2500 births depending on the population [1,2]. The majority of cleft lip and palate cases occur as a non-syndromic isolated phenotype with complex disease etiology, while about 30% of cases are syndromic occurring in association with other mendelian phenotypes [1,2]. Previous studies have shown that both genetic and environmental factors contribute to the cause of orofacial clefts making it difficult to identify the main etiology in many cases; however, it was also shown that many of the syndromic cases of CL/P were due to chromosomal abnormalities and/or monogenic causes [3].

Currently, more than 500 syndromes have been found to be associated with syndromic CL/P, with Van der Woude syndrome (VWS, OMIM 119300) being the most common, accounting for about 2% of all CL/P cases [4,5]. VWS is dominantly inherited and characterized by orofacial clefts with additional phenotypes including lower lip pits and hypodontia in some cases [5,6]. Like VWS, Popliteal Pterygium Syndrome (PPS, OMIM 119500) is an autosomal dominant condition also characterized by orofacial clefts, lower lip pits with additional features including genital and skeletal anomalies, skin webbing, and syndactyly of toes or fingers [5,6]. PPS is considered a more serious form of VWS with popliteal webbing, CL/P, and syndactyly used as key clinical differentiators occurring in more than half of all PPS cases [4,7]. Both VWS and PPS are very rare occurring in about 1 in 35,000 and 1 in 300,000 births respectively, and both are reported to be caused by mutations in the Interferon Regulatory Factor 6 gene (*IRF6*, OMIM 607199) [4,5]. 

Structural variants (SVs) are a class of genomic variation that are larger than 50 base pairs in size and account for about 1% of differences in terms of base content across human populations [8]. SVs may be simple deletions and duplications (together referred to as copy number variants (CNVs)) or complex events, including inversions, insertions, and translocations, all of which may either be balanced or unbalanced and/or co-occur with CNVs, in some cases involving three or more breakpoints [9]. Like other classes of genomic variation, SVs can be either benign or disease-causing, where they lead to gene disruption or gene dosage alterations [8]. Previous studies have reported complex SVs in patients with different Mendelian disorders; for instance, an inverted triplicated segment within a duplication was found at MECP2 and PLP1 gene loci in patients with Lubs syndrome and Pelizaeus-Merzbacher disease [9,10]. Additionally, recent studies reported the association of complex structural variants with neuropsychiatric conditions and ASD [9]. Despite these discoveries, the true prevalence of complex SVs in Mendelian disorders remain poorly understood due to the analytical and technical challenges of both discovering and interpreting such variants from short read data analysis pipelines, which are typically used in both whole exome (WES) and whole genome sequencing (WGS) today [9].

Recent improvements in variant identification tools, and in particular the development of more sensitive and specific algorithms for SV discovery, have aided the effort to identify and understand complex structural variations from short or long DNA fragments [11,12]. Once identified, de novo assembly of genomic regions of interest may help resolve difficult regions and understand their role of complex SVs in Mendelian disease [13,14].

Here, we describe the use of WGS on a family with both VWS and PPS who were initially negative by clinical WES, but in whom we use WGS with combined SV tools to identify and validate a novel complex intrachromosomal rearrangement on chromosome 1 disrupting the IRF6 protein. We believe such approaches could be generalized to other pediatric syndromic disorders that are exome-negative but for which WGS data can be generated and where orthogonal SV detection approaches may help identify genetic etiology.

## 2. Detailed Case Description

The index patient is a female 2nd child to non-consanguineous parents. She presented to the Plastics and Craniofacial clinic at 17 months of age and was diagnosed with Popliteal Pterygium syndrome (PPS). She had bilateral cleft lip (complete on the left, incomplete on the right) and cleft palate (both were repaired elsewhere) with large residual anterior fistula (Figure 1B). She also had lower lip pits and cysts on both sides, right popliteal pterygium, and bilateral complete simple syndactyly of second and third webspace of the hands (only nail plates fused were on the right-hand 3rd–4th digits) (Figure 1C–E). She was otherwise developmentally normal, had normal intelligence, normal cardiovascular and respiratory system, with no other anomalies or medical conditions noted. The older sister presented to our institution at 38 months of age with bilateral cleft lip and cleft palate (both repaired elsewhere) with large residual anterior fistula and had lower lip pits and cysts on both sides consistent with the diagnosis of van der woude syndrome (VWS) (Figure 1F). Similarly, the father -39 years old- also showed clinical features of VWS, having bilateral cleft lip (repaired elsewhere), unrepaired cleft of the entire secondary palate, and lower lip pits and cysts (Figure 1G). Both the father and the older sister were otherwise normal, with no other anomalies or medical conditions noted. Sanger sequencing of the *IRF6* gene revealed no candidate pathogenic variants, prompting enrollment into the Qatari Mendelian Disease Program, where whole genome sequencing (WGS) was performed for the entire family.

## 3. Results

The family was first referred with a suspicion of PPS and VWS, however routine pathology investigation, including re-sequencing of known gene panels and microarray testing, found no pathogenic variants in *IRF6*—a known candidate gene causing both conditions–segregating with disease in this family. The family was thus enrolled for WGS as part of the Qatar Mendelian Disease Program [15], to identify a genetic etiology for their clinical phenotype. All family members were enrolled by informed consent, and their genomes were sequenced to a minimum depth of 30×, and data was processed to identify rare, putatively pathogenic variants segregating with disease (for details see Appendix A). The WGS analysis revealed no immediate candidate pathogenic variants in the sub−50 bp range (single nucleotide variants or insertions/deletions (INDELS)). We therefore employed an in-house structural variant discovery pipeline (see Appendix A), to look for possible chromosomal abnormalities that could be causing the phenotype.

Among 136 deletion, duplication, and inversion events shared by affected family members (Figure 2), we detected a copy-neutral rearrangement segregating in affected family members on the long arm of chromosome 1 (1q32.2) that appeared to overlap the *IRF6* gene. This event was captured by our analysis pipeline as two consecutive deletions, 177 kb and 251 kb, that were both encompassed by a larger 429 kb duplication event (Figure 3). To further resolve the possible rearrangement event, we selected all reads within 500 kb window upstream and downstream of the putative breakpoints and proceeded with de novo assembly of short-read data at this locus. This approach revealed a rearranged allele in which the sequence of the two candidate ‘deleted’ segments originally detected were in fact reversed in order along the same chromosome, i.e., a deletion of the upstream segment and re-insertion of this first segment downstream of the second segment. Notably, the breakpoints identified from the de novo assembly suggested that exon 6 of the *IRF6* gene was disrupted (Figure 4A). By comparison to publicly available as well as internal Qatari structural variation data [16], this variant appeared to be novel, and was shared among the 3 affected family members, and absent from the unaffected mother and sibling, and thus consistent with autosomal dominant disease etiology.

We then proceeded to validate this rearrangement in the lab. To do this, we designed 3 sets of primers around the rearranged breakpoints (Figure A1). We amplified these genomic segments using PCR, and found the reference allele in all family members. While unaffected members were homozygous for the reference allele, the 3 affected members were heterozygous for it, alongside the rearranged allele as predicted from the in silico analysis (Figure 4A,B).

## 4. Discussion and Conclusions

In this paper, we present a two-generation family with an intrafamilial phenotypic variability who were found to have a complex rearrangement event on chromosome 1 disrupting the continuity of the *IRF6* gene. The index patient was diagnosed with popliteal pterygium syndrome, while the father and the sister showed clinical features of van der woude syndrome, both of which are caused by mutations in the *IRF6* gene.

Both popliteal pterygium and van der woude syndromes are dominantly inherited and mainly characterized by orofacial clefts. In popliteal pterygium syndrome, the most common feature present in more than 90% of cases is cleft palate with/without cleft lip and it includes additional cutaneous, genital, and musculoskeletal phenotypes such as popliteal skin webbing occurring in 58% of cases, genital anomalies in 37%, syndactyly in 50% and nail anomalies in 33% of cases, and any three of these phenotypes must be present for the diagnosis of PPS [17]. On the other hand, in addition to the cleft lip and/or palate, lower lip pits must be present in order to be diagnosed with van der woude syndrome [4].

Previously, Tan et al. reported a de novo 2.3 Mb microdeletion of 1q32.2 region involving the *IRF6* gene in a patient diagnosed with van der woude syndrome [6]. Upon searching the literature, no cases with complex structural variants involving the *IRF6* gene locus were found the variant in this family was a highly complex rearrangement that did not affect exonic sequence, and would therefore be missed using routine targeted and/or exome sequencing approaches. While it is known that families with IRF6 mutations can have significant inter- and intra- familial phenotypic variability, such cases are normally seen in families with point mutations in *IRF6*, no previous cases were reported with such complex intrachromosomal rearrangement event [18]. Further, the wide phenotype diversity seen in the previously described and current cases, supports the idea that the two syndromes, VWS and PPS, represent two ends of a phenotypic spectrum of a single condition rather than two separate disorders [18].

The use of WGS offers the ability to detect a multitude of genomic variants ranging from simple base-substitutions to complex rearrangements, including copy number changes, as well as copy-neutral inversions and translocations [19]. However, a major challenge remains accurately analyzing, filtering, and interpreting WGS data to identify such variants [19]. In this study, we identified variants on chromosome 1 that by traditional tools suggested 2 successive deletions overlapping a duplication. This prompted further analysis to resolve this variant. We therefore leverage de novo local genome assembly as a powerful approach to reconstruct the allele from the ground up. This approach revealed a complex intra-chromosomal translocation, where one genomic segment was removed and re-inserted downstream of its neighboring segment on the 1q32.2 region. This rearrangement affecting exon 6, disrupting the continuity of the IRF6 gene, and causing the dominantly inherited phenotypes observed in this family.

In the human genome, structural variants (SVs) are considered a significant source of variation, and it was shown that complex structural variants, representing 2% of SVs, are more abundant than what have been previously thought, playing an important role in Mendelian diseases [9]. However, due to the challenge in identifying and interpreting such complex SVs, they are typically missed or under-reported during genomic analyses. This familial case study demonstrates the potential of using de novo assembly from short-read WGS data and the use of several structural variant detection tools to identify such rare and complex intrachromosomal rearrangements, which could help with solving the missing heritability in a subset of patients with rare diseases in clinical settings.

## Figures and Tables

**Figure 1 genes-14-00849-f001:**
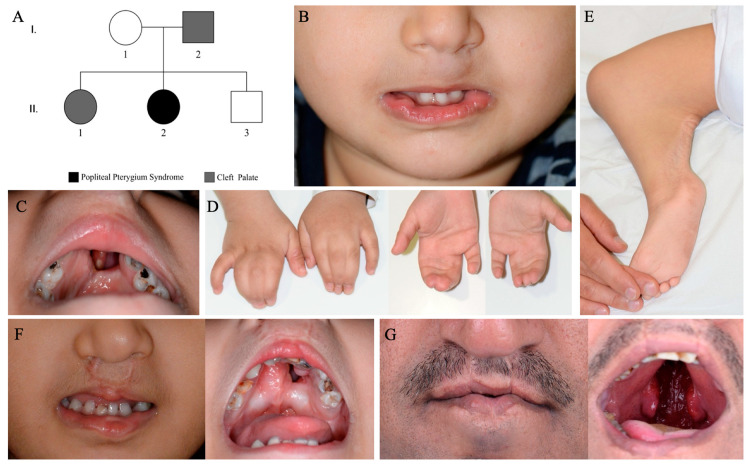
Family pedigree and clinical phenotypes in proband and affected family members. (**A**) Pedigree of family members shows the proband (II-2) with popliteal pterygium syndrome (PPS), father (I-2) and sister (II-1) with van der woude syndrome (VWS); (**B**) Proband with repaired bilateral cleft lip and lower lip pits and cysts on both sides; (**C**) Proband with cleft palate; (**D**) Proband with bilateral complete, simple syndactyly of second and third webspace of the hands; (**E**) Proband with right popliteal pterygium; (**F**) Sister with repaired bilateral cleft lip and palate and lower lip pits and cysts on both sides; (**G**) Father with repaired bilateral cleft lip, unrepaired cleft of the entire secondary palate, and lower lip pits and cysts.

**Figure 2 genes-14-00849-f002:**
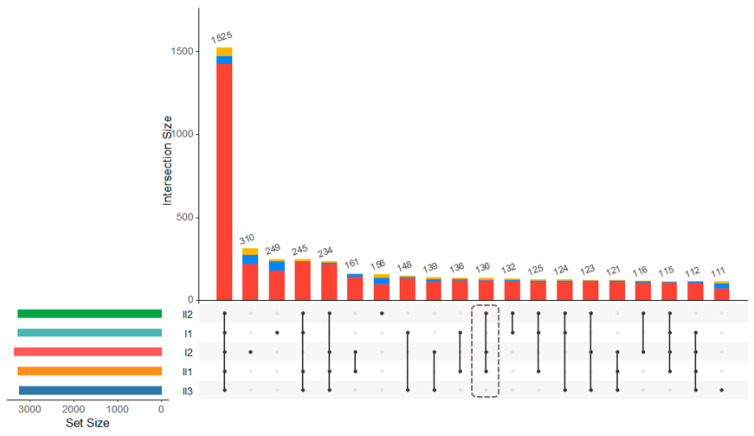
UpSet plot of variant intersection between the family members. Dotted line showing the shared 136 events in affected family members.

**Figure 3 genes-14-00849-f003:**
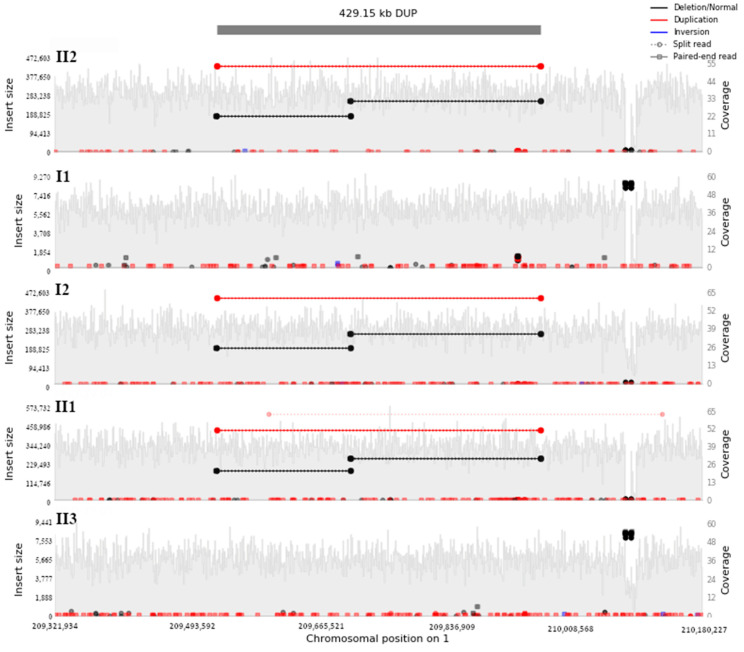
Visualization of the rearranged allele appeared as two consecutive deletions encompassed by a bigger duplication at the same region on chromosome 1 segregating in all affected family members.

**Figure 4 genes-14-00849-f004:**
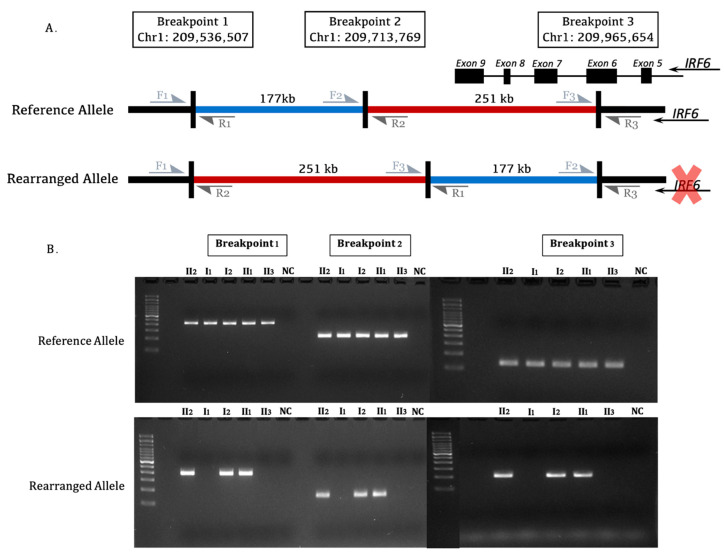
(**A**) Diagram showing the rearrangement event occurring in affected family members where the red and blue regions are switched disturbing the continuity of *IRF6* exon 6; (**B**) Gel electrophoresis of PCR products (100 bp ladder used) from the reference and rearranged alleles. All family members had the reference allele, while only the affected family members had the rearranged allele. F1/R1, F2/R2, and F3/R3 are the 3 primer sets (Forward and reverse) designed around the breakpoints.

## Data Availability

To respect patients’ privacy and confidentiality, their individual genome data cannot be shared.

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
