# Peer review of "A Complex Intrachromosomal Rearrangement Disrupting IRF6 in a Family with Popliteal Pterygium and Van der Woude Syndromes"

_genes, 2023, doi:10.3390/genes14040849_

Round 1
Reviewer 1 Report
This article reports a case of a two-generation family with popliteal pterygium syndrome and discovers and validates a novel, copy-neutral, 429kb complex intra-chromosomal rearrangement in IRF6 gene. These methods identified are valuable for future investigations. There are some concerns to be addressed.
1. Although Gel electrophoresis of PCR has been confirmed, Sanger sequencing across breakpoints may be more specific for SV, which needs to be explained why the PCR is used.
2. Image resolution of the Figure 1G is low and not clear enough.
3. The Figure 2 should show note the dotted line in line 145.
4. “SVABA” should be changed to “SvABA” in the manuscript.
Reviewer 2 Report
In this manuscript, the authors report a valuable case of a two-generation family with Popliteal Pterygium syndrome and Van der Woude syndrome. Using whole genome sequencing (WGS) followed by local de novo assembly, a novel intra-chromosomal rearrangement disrupting the IRF6 gene is identified. The manuscript is well organized and written. The results are clearly presented. To improve the paper, please find my suggestions and comments.
1. Line 115, “routine pathology investigation found no pathogenic variants in IRF6”. What methods were used for the routine pathology investigation? “…re-sequencing of known gene panels or microarray testing” is mentioned in Abstract, but it is not addressed in the main text.
2. Figure 2, it would be helpful to highlight or mark the 136 events shared by affected family members, so readers can note the intersection easily.
3. Figure 4, the PCR bands from the rearranged allele are clear. It would be more convincing if the PCR products from the rearranged allele can be analyzed by Sanger sequencing, visualizing the sequences of the breakpoints predicted from the in silico analysis.
4. How does Breakpoint 3 affect the expression of IRF6 (mRNA and protein)? Why was it not detected by re-sequencing of known gene panels or microarray testing, even if this breakpoint is located within Exon 6?
5. Appendix A, Primer design. The sequences of the primers used should be provided.
